# Controlling oncogenic KRAS signaling pathways with a Palladium-responsive peptide

Soraya Learte-Aymamí[1], Pau Martin-Malpartida[2], Lorena Roldán-Martín[3], Giuseppe Sciortino[3,4], José R. Couceiro[1], Jean-Didier Maréchal[3], Maria J. Macias[2,5], José L. Mascareñas ![ORCID] [1✉] & M. Eugenio Vázquez ![ORCID] [1✉]

RAS oncoproteins are molecular switches associated with critical signaling pathways that regulate cell proliferation and differentiation. Mutations in the RAS family, mainly in the KRAS isoform, are responsible for some of the deadliest cancers, which has made this protein a major target in biomedical research. Here we demonstrate that a designed bis-histidine peptide derived from the αH helix of the cofactor SOS1 binds to KRAS with high affinity upon coordination to Pd(II). NMR spectroscopy and MD studies demonstrate that Pd(II) has a nucleating effect that facilitates the access to the bioactive α-helical conformation. The binding can be suppressed by an external metal chelator and recovered again by the addition of more Pd(II), making this system the first switchable KRAS binder, and demonstrates that folding-upon-binding mechanisms can operate in metal-nucleated peptides. In vitro experiments show that the metallopeptide can efficiently internalize into living cells and inhibit the MAPK kinase cascade.

[1] Centro Singular de Investigación en Química Biolóxica e Materiais Moleculares (CiQUS), Departamento de Química Orgánica, Universidade de Santiago de Compostela, Santiago de Compostela 15705, Spain. [2] Institute for Research in Biomedicine (IRB Barcelona), The Barcelona Institute of Science and Technology, Barcelona 08028, Spain. [3] Insilichem, Departament de Química, Universitat Autònoma de Barcelona, Cerdanyola 08193, Spain. [4] Institute of Chemical Research of Catalonia (ICIQ), The Barcelona Institute of Science and Technology, Tarragona 43007, Spain. [5] Institució Catalana de Recerca i Estudis Avançats (ICREA), Passeig Lluís Companys 23, Barcelona 08010, Spain. ✉email: joseluis.mascarenas@usc.es; eugenio.vazquez@usc.es

RAS proteins are oncogenic supervillains mutated in almost half of all cancers[1]. Many of these mutations drive some of the most lethal cancer types, including about 95% of pancreatic, 45% of colorectal cancers, and 33% of lung adenocarcinomas[2]. More specifically, KRAS, the most frequently mutated isoform, is responsible for most RAS cancers and generates highly aggressive and life-threatening phenotypes[3]. These features have made KRAS a key target in the search for new antitumor agents for over three decades[4,5]. Physiologically, RAS proteins are binary molecular switches that oscillate between an inactive, GDP-bound, state and an active RAS-GTP complex, aided by specific guanine nucleotide exchange factor proteins (GEFs), such as SOS1, which stimulate the GDP/GTP exchange and activation, and GTPase-activating proteins (GAPs) that catalyze the hydrolysis of GTP and inactivation in the RAS-GDP state. Certain mutations, such as KRAS[G12C] or KRAS[G12V], compromise the GTPase activity of RAS proteins, locking them in their active state, thereby inducing the aberrant activation of several signaling pathways that ultimately lead to uncontrolled cell growth and proliferation, invasiveness, angiogenesis, and metastasis[6,7]. Activation of RAS-GDP by SOS1 is induced by the insertion of the αH helix of the exchange factor, which opens up the GTP binding pocket and reduces the affinity for the trinucleotide[8]. It has been shown, that conformationally restricted peptidomimetics of the SOS1 αH helix featuring helix-capping modifications[9,10], hydrocarbon staples[11], and SOS1 protein mimics[12], bind KRAS with high affinity, and competitively inhibit RAS activation by SOS1.

In contrast to these constitutively active inhibitors, locked into stable α-helical conformations by permanent linkages, stimuli-responsive peptides that change their affinity in response to an external input could offer new opportunities in basic research, as well as for spatiotemporal controlled therapeutic applications. Thus, going beyond the classic metal-stabilized α-helices[13–16], and considering our previous studies in the field of DNA binding[17–20], we hypothesized that metal chelation could be applied for the development of stimuli-responsive KRAS peptide inhibitors[21,22]. Indeed, here we show that the peptide **αH-His₂**, engineered to include two *i,i + 4* His chelating residues in a short fragment of the SOS1 αH helix, binds KRAS only after addition of a Pd(II) source. We also show that such coordination is reversible, and that the metallopeptide **αH-His₂**[Pd] is capable of inhibiting KRAS-activated pathways in live cells. Moreover, we also found that complete preorganization of the peptide α-helix is not required for efficient binding and inhibition, and that **αH-His₂**[Pd] operates through a coupled folding-upon-binding mechanism, typical of Intrinsically Disordered Proteins.

## Results

**Design and synthesis of the metal-responsive KRAS peptide inhibitor.** Based on the reported examples of SOS1 αH peptidomimetics[11], as well as on the structural details of the SOS1 αH inserted between the switch regions of KRAS (PDB codes 1NVW and 1BKD)[8,23], we chose the fragment comprising SOS1 residues Phe929 to Asn944, which spans the full αH helix for our experiments. Inspection of the hypothetical αH/RAS complex revealed that positions Tyr933 and Ile937 would be located in the solvent-exposed face of such αH helix fragment, and distanced so that their side chains, when replaced by His residues, would be in register and appropriately oriented to chelate a Pd(II) center (Fig. 1). In addition to the Tyr933His and Ile937His mutations, the αH sequence in our final design included two additional Arg residues in its N-terminus, for increased solubility and cellular uptake, and the sequence was capped with 6-carboxytetramethylrhodamine (TMR) for quantification and fluorescence monitoring (Fig. 1, **αH-His₂**). The peptides

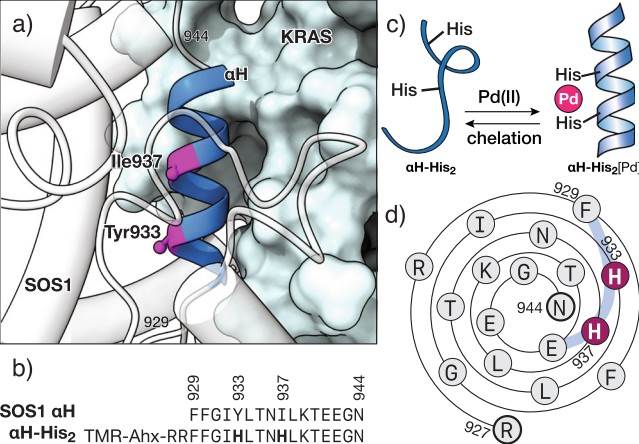

**Fig. 1 Design of the metal responsive KRAS peptide inhibitor. a** Detail of the αH helix of SOS1 complexed with RAS (PDB code 1BKD) indicating the residues Tyr933 and Ile937 that are replaced by the metal chelating His residues; the structure of SOS1 is shown in white. **b** Sequence of the natural αH helix of SOS1 and the mutant peptide **αH-His₂** used in this study, highlighting the two engineered His residues. **c** Conformational control by Pd(II) chelation. **d** Wenxiang diagram of the **αH-His₂**[79,80], indicating the residues aligned in the outer face of the αH helix, including the two histidine residues in consecutive turns of the α-helix.

were synthesized following standard Fmoc solid-phase peptide synthesis methods on an automated microwave synthesizer[24,25].

**Addition of Pd(II) increases the α-helical content of the peptide αH-His₂.** Mixing the peptide **αH-His₂** with 1 equiv. of *cis*-PdCl₂(en) gave the expected metallopeptide, **αH-His₂**[Pd] from now on, as confirmed by NMR by following the chemical shift changes observed at the His ring protons, and by MS (see below and Fig. S2). Circular dichroism studies showed that metal coordination moderately increases the α-helical content of the peptide. Thus, a solution of **αH-His₂** displayed the typical CD signature of a random coil peptide and a low-intensity band at 222 nm ($\approx -2200$ deg cm² dmol⁻¹); addition of one equivalent of *cis*-PdCl₂(en) induced a modest increase in the negative intensity of this band ($\approx -5\,700$ deg cm² dmol⁻¹) that is consistent with a relatively small α-helical content of about 22% (Fig. 2, left; Fig. S3)[26]. Earlier experiments in our laboratory show that square planar *cis*-Pd(II) complexes display the ideal coordination properties to promote the effective folding of the helical conformation while other geometry (*trans*-[Pd(PPh₃)₂Cl₂]), or other metal ions, such as Cu(II), Co(II), Ni(II), Fe(II), do not stabilize the active conformation[22]. Interestingly, the addition of a small excess of the Pd(II) chelator diethyldithiocarbamate (DEDTC), gave rise to a marked reduction in the helicity, supporting the key role of the metal in stabilizing the α-helical conformation. Subsequent addition of *cis*-PdCl₂(en) led to a recovery of the helicity, thus confirming that the system can be easily switched between the metallo-stapled and the metal-free forms (Fig. 2, right).

**NMR spectroscopy demonstrates that the αH-His₂ core sequence partially folds into an α-helical conformation upon metal chelation.** We characterized in detail the conformational effects of the Pd(II) coordination by NMR spectroscopy. All peptide residues were unambiguously assigned combining spin system identification in the TOCSY experiment and sequential assignment via NOEs (SI, pages S7-S11). In agreement with the CD studies, we found that the free peptide **αH-His₂** is mostly unstructured in solution, with a certain propensity to populate an

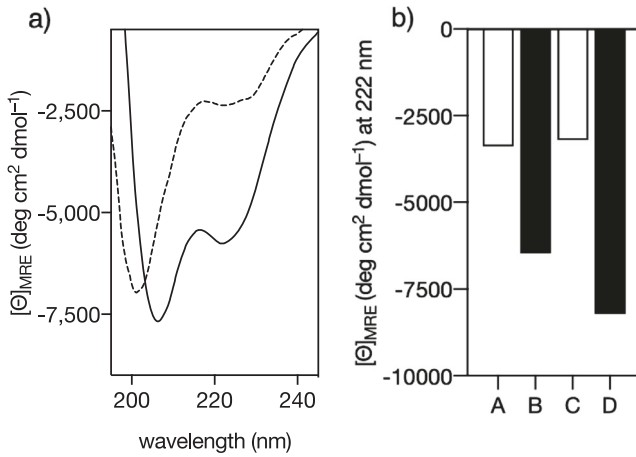

**Fig. 2 Circular Dichroism studies show that the α-helical content of the peptide αH-His₂ can be switched. a** Circular dichroism of a 20 µM solution of **αH-His₂** (dashed line) and the spectrum 30 min after addition of 1 eq. of *cis*-PdCl₂(en); **b** CD signal at 222 nm for: A 20 µM solution of **αH-His₂**; B same solution after addition of Pd(II) and formation of the **αH-His₂[Pd]** metallopeptide; C solution in B after addition of 5 eq. of DEDTC; D solution in C after addition of 20 eq. of *cis*-PdCl₂(en). The experiments were carried out in 10 mM phosphate buffer pH 7.5, 100 mM of NaCl and 10% TFE, 25 °C. Mean residue molar ellipticity ([Θ]MRE) was calculated considering an 18-mer.

α-helical conformation in the central part of the sequence, as deduced from the pattern of NH-NH NOEs observed for residues 929–940. The addition of *cis*-PdCl₂(en) (up to 3 equiv.) led to a significant folding of the peptide chain, as deduced from several long-range NOEs between the side chains of Phe929, Ile932, and Leu934. These NOEs were absent in the free peptide and suggest that in addition to locking the α-helical conformation in the vicinity of the coordinating residues His933 and His937—from residues Gly931 to Leu938—metal chelation also promotes the formation of a N-terminal helical conformation from residues Arg928 to Phe930 (Fig. 3, Methods).

**Molecular modeling provides further insight into the conformational dynamics and metal coordination of the peptide αH-His₂.** To understand the conformational effect of the metal chelation in the α-helical conformation, we ran a series of 2 µs Gaussian accelerated molecular dynamics (GaMD) simulations of the peptide **αH-His₂** in its free form and coordinated with Pd(II). For the latter, the starting model was generated by performing protein-ligand docking of the Pd(II) metal center to the peptide using approaches updated for metal ligands[27–30]. The simulations show in all cases a largely disordered *N*-terminal region (from residues Arg927 to Ile932). In the case of the metal-free peptide, followed by an unstable C-terminal α-helix that undergoes multiple folding/unfolding cycles until, at the end of the simulation, the conformational sampling leads to a long-lasting ensemble of unfolded structures (Supplementary Information, Figs. S4a and S5a). This indicates that the peptide has some intrinsic helical propensity, albeit insufficient to reach a stable fold. This result is supported by the experimental evidence that points to a largely disordered state in this metal-free form. In contrast, the simulation of the metallopeptide **αH-His₂[Pd]** with *N*τ coordination in both His led to a more stable fold with substantial α-helical propensity throughout the 2 µs simulation. The core region from residues His933 to Glu941, where the coordination occurs, rarely deviates from a canonical α-helix and when it does, it adopts other organized secondary structures (e.g., 3₁₀ helix). No

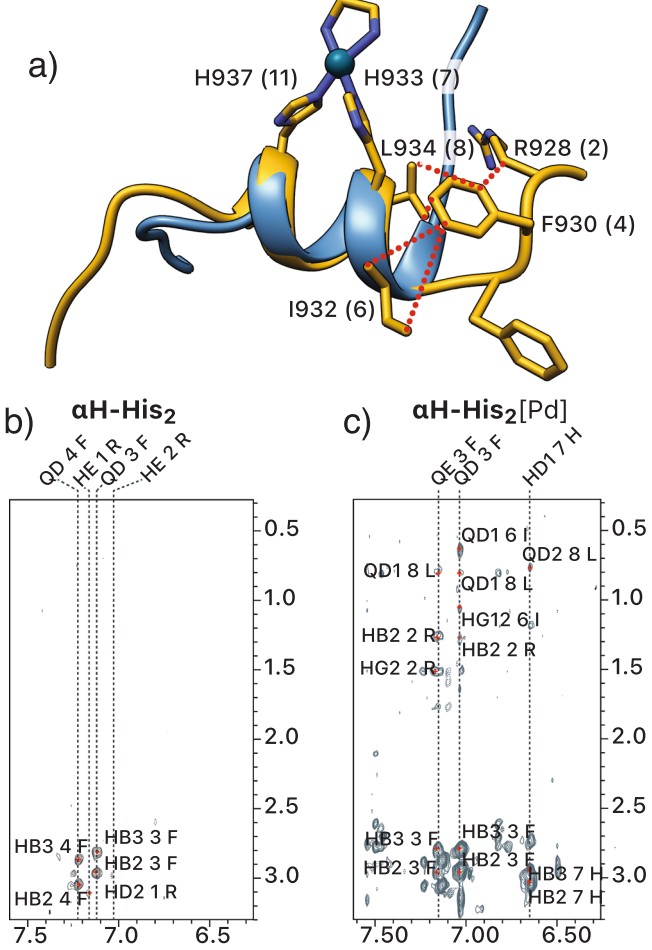

**Fig. 3 NMR spectroscopy and computational studies confirm the nucleation of the α-helix in αH-His₂[Pd]. a** Experimental NMR conformation in solution (lowest energy) in yellow; NOEs are indicated as dotted lines superimposed with a representative conformation of the MD simulation (in blue). **b, c** Aromatic-side chain regions of the NOESY spectra. **b** native peptide. **c** in the presence of *cis*-PdCl₂(en). Residues participating in NOEs are labeled. In the left panel only intra-residue NOEs are detected whereas in the right panel, both intra and inter residue NOEs are observed, indicating the presence of a helical conformation upon Pd binding. PDB structures available as Supplementary Data 1.

transition to a random coil is observed (Figs. S4b and S5b). A comparison between the MD most sampled conformation of the **αH-His₂[Pd]** and the lowest energy NMR conformer in solution shows an excellent agreement (Fig. 3). An additional control MD run with an alternative coordination mode (*N*τ-His933/*N*π-His937) showed a marginally stable helical fold with frequent deviations of the ideal α-helix geometry and even completely unfolded structures. Such poor structuring effect is clearly shown by the representative structure of the most populated sub-ensemble, which shows a kink in the α-helix at the coordination site (Figs. S4c and S5c). Altogether, the computational study provides an excellent match with the spectroscopic data, confirming the enhancement of α-helical conformation through dual coordination of the His residues through their *N*τ atoms.

**The metallopeptide αH-His₂[Pd] binds KRAS^wt.** The above studies demonstrate that coordination to the Pd(II) modulates the conformation of the peptide **αH-His₂** in solution, nucleating the formation of an incipient α-helix. We anticipated that this metal-induced folding might facilitate the interaction of the peptide

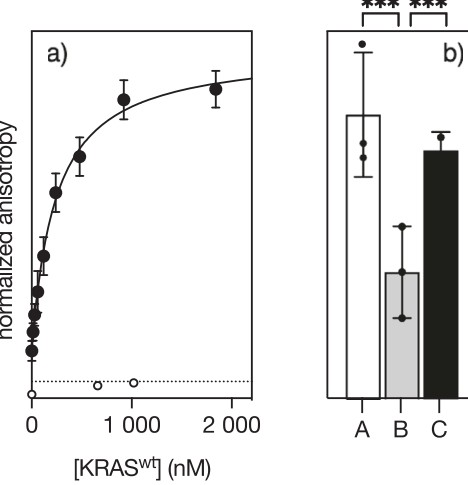

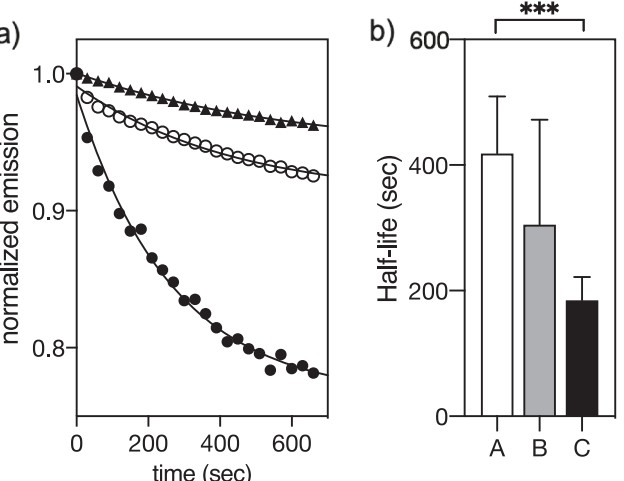

**Fig. 4 Binding of αH-His₂[Pd] to KRAS, and RAS-binding switching studied by fluorescence anisotropy. a** Normalized emission anisotropy at 559 nm of a 15 nM solution of **αH-His₂[Pd]** with increasing concentrations of KRAS$^{wt}$. The best fit to a 1:1 binding model is also shown[43,44]; **b** anisotropy values for: A, a saturated **αH-His₂[Pd]**/KRAS complex; B, same solution after addition of 50 eq. of DEDTC; C, solution in B after addition of 50 eq. of cis-Cl₂Pd(en). Data are mean ± SEM for experiments performed in technical triplicate and are representative of at least three biological replicates performed with independent preparations of recombinant KRAS$^{wt}$. All experiments carried out at 25 °C. Data were subjected to a One-Way ANOVA statistical analysis, ***$p < 0.001$.

**Fig. 5 Binding of the metallopeptide αH-His₂[Pd] to KRAS inhibits the nucleotide release process.** Preformed **αH-His₂[Pd]** increases the displacement of mantGTP bound to KRAS$^{wt}$. **a** normalized emission at 430 nm. Black triangles: intrinsic nucleotide displacement curve of a 1 μM solution of the preformed KRAS$^{wt}$/mantGTP complex; white circles, same solution in the presence of 10 μM **αH-His₂**; black circles: same KRAS$^{wt}$/mantGTP solution in the presence of 10 μM **αH-His₂[Pd]**. **b** half-life of the kinetic traces shown on the left. A intrinsic KRAS$^{wt}$/mantGTP complex; B with **αH-His₂**; C with **αH-His₂[Pd]**. Data were subjected to a One-Way ANOVA statistical analysis, ***$p < 0.001$.

with KRAS. Indeed, mixing a 15 nM solution of the unstructured **αH-His₂** in absence of Pd(II) with increasing concentrations of wild-type KRAS$^{wt}$ did not produce any increase in the fluorescence anisotropy of the TMR fluorophore at 559 nm, indicating that the apopeptide, **αH-His₂**, has low binding affinity for KRAS$^{wt}$. In contrast, when the titration was repeated with a preformed palladium/peptide complex, **αH-His₂[Pd]**, we observed a marked increase in the anisotropy, which is consistent with binding to KRAS$^{wt}$ and formation of a larger complex with slower rotational diffusion. The increase in anisotropy could be fitted to a simple 1:1 binding model with an approximate dissociation constant of 240 nM (Fig. 4a). Similar titration profiles were obtained when **αH-His₂[Pd]** was incubated with the oncogenic mutants KRAS$^{G12C}$ and KRAS$^{G12V}$, which displayed also tight binding with dissociation constants of 345 nM and 294 nM, respectively (Fig. S6). Importantly, the addition of DEDTC to the solution of the complex **αH-His₂[Pd]**/KRAS$^{wt}$[18,31,32], resulted in a marked decrease of the TMR anisotropy, indicating the disassembly of the metallopeptide **αH-His₂[Pd]**, and disengagement of **αH-His₂** from KRAS$^{wt}$. Furthermore, the addition of fresh cis-PdCl₂(en) to that solution restored the original anisotropy, indicating that the released peptide **αH-His₂** was still able to chelate Pd(II) metal center and to bind once again as **αH-His₂[Pd]** to KRAS$^{wt}$ (Fig. 4b), thereby confirming the reversibility of the system.

Taken together, these data confirm that the palladium clip is working as a facilitator of the helix folding rather than as a traditional staple in which the covalent link fixes the bioactive α-helical conformation[33]. In other words, the coordination to the Pd(II) appears to provide just enough thermodynamic stabilization for the peptide to display folding-upon-binding properties such as those typically observed for Intrinsically Disordered Proteins[34,35]. A similar effect has been described for short trigger sequences that mediate the folding of coiled-coil peptides and allow the peptides to surpass a critical threshold of stability[36,37].

These results are also consistent with the region of the metal clip being highly structured, as shown by NMR and MD studies, despite the low total helical content of the metallopeptide as measured by CD.

**The metallopeptide αH-His₂[Pd] inhibits KRAS$^{wt}$ GTP exchange.** Having confirmed the high affinity of the complex **αH-His₂[Pd]** for KRAS$^{wt}$, we studied its effect in the nucleotide release process associated with biological inactivation of this protein[38]. Therefore, we first measured the intrinsic nucleotide release rate by monitoring the fluorescence emission of the complex of KRAS$^{wt}$ with the GTP analog mantGTP, 2′-/3′-O-(N-methylanthraniloyl) guanosine-5′-triphosphate, which displays low fluorescence in solution, but becomes strongly emissive when associated with KRAS$^{wt}$[21,22]. As expected, the KRAS$^{wt}$/mantGPT complex was very stable, and fluorescent mantGTP was released very slowly, with a half-life of about 7 min (Fig. 5). The half-life of this complex was somewhat decreased by the presence of the free peptide **αH-His₂** (Fig. 5 right, $t_{1/2} \approx 5$ min), but the addition of the preformed metallopeptide **αH-His₂[Pd]** drastically increased the rate of nucleotide release, so that the measured half-time of the KRAS$^{wt}$/mantGPT complex was reduced to about 3 min under the same conditions (Fig. 5, right).

**The metallopeptide αH-His₂[Pd] inhibits the ERK1/2 signaling pathway in cells.** The coordinative basis of our stapled peptide raised the question of whether it could be used in the complex cellular environment. Thus, to assess its capacity to antagonize the KRAS activation in living mammalian cells, we first monitored its cellular uptake by fluorescence microscopy. Incubation of lung carcinoma A549 cells with **αH-His₂** revealed a rather poor internalization (Fig. 6a), but when this peptide was previously incubated with cis-PdCl₂(en), the resulting metallopeptide, **αH-His₂[Pd]**, showed an intense intracellular fluorescence signal (Fig. 6b). Therefore, the palladium clip facilitates the internalization[39]. We then analyzed the activation of the MAPK

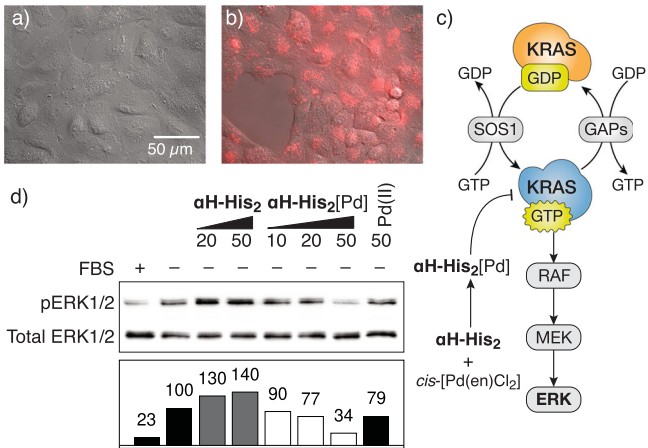

**Fig. 6 αH-His₂[Pd] is internalized and promotes the inhibition of the MAPK kinase cascade. a, b** Effect of Pd(II) coordination in cellular internalization (A549 cells). **a** Incubation with **αH-His₂** (10 μM, 30 min at 37 °C); **b** incubation with the preformed **αH-His₂[Pd]** complex, prepared by mixing an equimolar mixture of **αH-His₂** and *cis*-PdCl₂(en) in water for 10 min (10 μM, 30 min at 37 °C). The cells were washed with PBS before being observed under the microscope. Incubations were made in Dulbecco´s modified Eagle medium (DMEM) completed with 5% of fetal bovine serum (FBS-DMEM). $\lambda_{exc} = 550$ nm, $\lambda_{em} = 590$–650 nm; **c** Scheme of the MAPK pathway and the proposed inhibition of the SOS1-mediated activation of KRAS by **αH-His₂[Pd]** that should lead to reduced phosphorylation of ERK; **d** Addition of *cis*-PdCl₂(en) to **αH-His₂** led to a palladium peptide complex that inhibits the RAF-MEK-ERK mitogen-activated protein kinase (MAPK) pathway downstream of KRAS in living cells, as deduced from the decrease in the amount of phosphorylated pERK. Serum-starved A549 cells were incubated either with **αH-His₂**, the preformed metallopeptide **αH-His₂[Pd]**, or with *cis*-PdCl₂(en) at the indicated doses for 4 h. Cellular lysates were subjected to Western blot analysis by using antibodies to phospho- and total ERK1/2. Densitometry measurement of the Western blot bands. The ratio between phospho-ERK1/2 and total ERK1/2 are indicated as fold change with respect to activation conditions (without FBS, lane 2). Activation percentage is shown above each column. Original WB and micrographies available as Supplementary Data 2.

RAF-MEK-ERK cascade, a well-documented KRAS effector pathway implicated in cell proliferation, survival, and differentiation (Fig. 6c)[40]. As expected, incubation of serum-starved A549 cells[41] with either the apopeptide **αH-His₂** or with *cis*-PdCl₂(en) failed to induce changes in the levels of ERK phosphorylation, as measured by blotting the cell lysates with pERK specific antibodies. In contrast, the addition of the preformed complex **αH-His₂[Pd]** to the cell culture led to a significant decrease in the levels of phosphorylated ERK (Fig. 6d), thus demonstrating that the palladapeptide is capable of inhibiting ERK phosphorylation pathways in a dose-responsive manner.

## Discussion

Peptidomimetics, particularly α-helix mimics, are promising platforms for the modulation of "undruggable" protein targets[42]. To date, the α-helical peptide inhibitors reported in the literature typically rely on the covalent linkages[43–45], including carbon staples[46–48], hydrogen bond surrogates[49–51], lactam bridges[52,53], or cysteine crosslinks[52,53], among others[54,55]. These covalent modifications afford peptides that are constitutively active, but having at hand stimuli-responsive peptidomimetics, which can be (de)activated at will, would open interesting opportunities both in

basic research and biomedicine. In this context, we envisioned the application of metal coordination to provide increased conformational stability and stimuli-responsive properties to a short α-helix peptidomimetic.

As model system, we chose the SOS1 αH helix/KRAS complex because of the intrinsic relevance of the KRAS GTPase in cancer as well as because there are clear precedents of short peptides derived from the αH helix that selectively bind to KRAS[21,22]. Furthermore, the available structural data of the SOS1/KRAS complex allowed us to rationally engineer the parent sequence to introduce the required residues for Pd(II) coordination in the exposed face of the αH helix in its complex with KRAS[54–56]. Thus, inspection of the reported structures of the SOS1/KRAS complexes suggested the replacement of Tyr933 and Ile937 with two His residues to create the metal-binding site[33]; these mutations, together with two additional Arg residues at the N-terminus ensured the solubility of the resulting peptide (RRFFGIHLTNHLKTEEGN). No further optimization of the sequence to increase its intrinsic α helical propensity was attempted. Indeed, the software AGADIR, accessible via http://agadir.crg.es/, predicts that the sequence of the **αH-His₂** peptide at 25 °C and pH 7 is only 12% helical[56–58].

Having at hand the desired peptide, **αH-His₂**, we used a combination of circular dichroism, NMR spectroscopy, and computational methods to study its coordination properties and the effect of Pd(II) binding in its folding. Both CD and NMR indicated that the free peptide **αH-His₂** was largely unstructured in solution and also showed a very modest stabilization of the α helical conformation in the presence of the metal ion. Thus, addition of Pd(II) results in a small increase in the negative molar ellipticity signal at 222 nm from $\approx$−2 200 to −5 700 deg cm² dmol⁻¹, and in the appearance of several long-range NOEs between the side chains of Phe929, Ile932, and Leu934. Molecular Dynamics studies confirm the $N\tau$ coordination of both His residues, and the partial folding of the peptide from residues Arg928 to Phe930. These effects were reversible, and addition of a Pd(II) chelator reduced the helical content of the peptide.

Despite the modest effect of Pd(II) chelation in the overall helical content of the **αH-His₂** peptide, Pd(II) coordination led to a dramatic change in the affinity for KRAS, so that while no significant binding was observed by fluorescence anisotropy when the free peptide **αH-His₂** was incubated with KRAS, titrations with the metallopeptide **αH-His₂[Pd]** resulted in clear binding isotherms with tight dissociation constants of 345 nM and 294 nM for KRAS$^{G12C}$ and KRAS$^{G12V}$, respectively. This effect is consistent with an induced folding-upon-binding mechanism facilitated by the nucleation of the bioactive α helical conformation when the Pd(II) coordinates the two His residues. Therefore, although the coordination of the metal ion does not provide enough energy to fold the metallopeptide **αH-His₂[Pd]** into an α-helix by itself, it is enough to do so in the presence of KRAS and the additional interactions with the protein that stabilize the bound helix. This behavior reproduces that of Intrinsically Disordered Proteins or regions of proteins, which are typically involved in protein-protein interactions regulating signaling pathways, that rely on their transition from a disordered to a folded state upon recognition of their physiological partners[58]. This mechanism allows high specificity, ultra-sensitivity, and switching behavior[59], which might also be desirable properties in designed peptide inhibitors. As expected from the initial coordination tests, addition of a Pd(II) chelator disassembles the metallopeptide **αH-His₂[Pd]** and takes apart the **αH-His₂[Pd]**/KRAS complex. Importantly, the formation of the **αH-His₂[Pd]**/KRAS complex has functional consequences, so that the inhibition of the GTPase activity of KRAS can be modulated in vitro by the presence of Pd(II).

Finally, the preformed metallopeptide **αH-His₂**[Pd] was internalized by A549 cells, as demonstrated by fluorescence microscopy, and the peptide inhibited the MAPK RAF-MEK-ERK cascade in a dose-responsive manner, as demonstrated by the measured decrease in the levels of phosphorylated ERK. To our knowledge, this is the first demonstration of a designed metallopeptide that can modulate a signaling pathway in living cells.

In this proof-of-concept work we challenge the current paradigm for the design of α-helix peptidomimetics, which seeks to maximize the α-helical content to avoid the entropic penalty associated with the folding of the peptide into its bioactive helical conformation. We demonstrate that such drastic preorganization is not required and that marginally stable helices can act as effective inhibitors through a folding-upon-binding mechanism, just like the natural Intrinsically Disordered Proteins that exploit this process to modulate protein activity. Moreover, we also report, for the first time, a strategy to obtain a switchable α-helical peptidomimetics that modify their folding and binding affinity to their targets in response to an external stimulus, namely, the presence of a Pd(II) metal ion. We applied this strategy for the design of a short peptide derived from the αH helix of the cofactor SOS1 that binds the oncogenic protein KRAS with high affinity upon coordination to Pd(II). NMR and MD studies provide a detailed atomistic picture of the nucleating effect of the metal center and support a folding-upon binding mechanism typical of Intrinsically Disordered Proteins[60]. In consonance with the metal-induced activation, the metal-free peptide, **αH-His₂**, is biologically inactive in cellular assays, but the metallopeptide **αH-His₂**[Pd] induced a significant inhibitory effect in KRAS-promoted pathways. Our approach uncovers new possibilities for the design of dynamic α-helix peptidomimetics that exploit the folding-upon-binding mechanism and can be extended to other biological targets.

## Methods

**Peptide synthesis**. The peptide **αH-His₂** was synthesized on a *Liberty Blue Lite* automatic microwave assisted peptide synthesizer from *CEM* Corporation, following the manufacturer's recommended procedures. The synthesis was on a 0.1 mmol scale using a 0.5 mmol/g load H-Rink amide *ChemMatrix* resin. Amino acids were coupled in 5-fold excess using DIC (*N,N*'-Diisopropylcarbodiimide) as activator, Oxime as base, and DMF as solvent. Couplings were conducted for 4 min at 90 °C. Deprotection of the temporal Fmoc protecting group was performed with 20% piperidine in DMF for 1 min at 75 °C. TMR was manually coupled to the N-terminus of the sequence by incubation of the resin-bound peptide with 3 eq. of 5-carboxytetramethylrhodamine (0.15 mmol, 64.5 mg), 3 eq. of HATU, and 5 eq. of DIEA 0.2 M in DMF for 60 min. Cleavage/deprotection was done by treating the resin-bound peptide for 2 h with 900 μL TFA, 50 μL CH₂Cl₂, 25 μL H₂O and 25 μL TIS (1 mL of cocktail/40 mg resin). The resin was filtered, and the filtrate was added onto ice-cold diethyl ether. After ~20 min, the precipitate was centrifuged and washed again with of ice-cold ether. The solid residue was dried under argon and redissolved in water/CH₃CN (1:1) for HPLC purification on a semipreparative *Agilent* 1100 series LC using a *Phenomenex Luna–C₁₈* (250 × 10 mm) reverse-phase column and a linear gradient (5 to 75% B over 40 min, 4 mL/min.; A: H₂O 0.1% TFA, B: CH₃CN 0.1% TFA). Collected fractions with pure products were freeze-dried to afford the desired peptide. The peptide was analyzed by analytical UHPLC-MS with an *Agilent* 1200 series LC/MS using an SB C₁₈ (1.8 μm, 2.1 × 50 mm) analytical column from *Phenomenex* with a linear gradient from 5 to 95% of solvent B for 20 min at a flow rate of 0.35 mL/min (A: water with 0.1% TFA, B: acetonitrile with 0.1% TFA). Compounds were detected by UV absorption at 222, 270, and 330 nm. Electrospray Ionization Mass Spectrometry (ESI/MS) was performed with an *Agilent* 6120 Quadrupole LC/MS model in positive scan mode using direct injection of the purified peptide solution into the MS detector.

**Circular Dichroism**. Circular Dichroism was measured on a *Jasco-715* coupled with a thermostat *Nestlab* RTE-111 using a 2 mm *Hellma* cuvette at 25 °C and the following settings: Acquisition range, 300-195 nm; band width, 2.0 nm; resolution, 0.2 nm; accumulation, 5 scans; sensitivity, 10 mdeg; response time, 0.25 s; speed, 100 nm/min. The spectra are the average of 5 scans and were processed using the "smooth" macro in the program *KaleidaGraph* (v 3.5 by Synergy Software).

**Fluorescence Anisotropy and switch experiments**. Measurements were made with a *Jobin-Yvon Fluoromax-3*, (DataMax 2.20), coupled to a *Wavelength Electronics LFI − 3751* temperature controller, and using a 1 mL *Hellma* micro cuvette. Settings: integration time, 2.0 s; excitation slit width, 5.0 nm; emission slit width, 20.0 nm; excitation wavelength, 559 nm; emission wavelength, 585 nm. Aliquots of a stock solution of protein were added onto 15 nM solutions of **αH-His₂** or **αH-His₂**[Pd] in Tris-HCl buffer 20 mM, pH 7.5, 100 mM NaCl, and the anisotropy was recorded after each addition. Experimental data correspond to the mean of three independent titration experiments and are representative of at least three biological replicates performed with independent preparations of recombinant proteins. All experiments carried out at 25 °C. Before use the corresponding purified protein, the glycerol was removed using a *Zeba* Spin Desalting Columns (*Thermo fisher*) that had been equilibrated with anisotropy buffer (Tris-HCl buffer 20 mM, pH 7.5, 100 mM NaCl). For the switch experiment, after completing the titration at saturating concentrations of the protein, 50 eq. of DEDTC (relative to the metallopeptide) was added to the solution and the anisotropy was recorded. After that, 50 eq. of *cis*-PdCl₂(en) was added (relative to the metallopeptide) and the anisotropy was recorded again.

**High-resolution NMR**. NMR experiments were recorded on a *Bruker* Avance III 600-MHz spectrometer (IRB Barcelona) equipped with a quadruple (¹H, ¹³C, ¹⁵N, ³¹P) resonance cryogenic probe head and a z-pulse field gradient unit at 298 K using a 1 mM solution of **αH-His₂** peptide—in the presence or absence of *cis*-PdCl₂(en)—1D proton spectra were recorded with a sweep width of 12000 Hz and 32 k data points. A total of 16 scans were accumulated with an acquisition time of 2.05 s. A Watergate w5 composite pulse was used to suppress the water signal. ¹H 2D-TOCSY and NOESY experiments were acquired in 90% H₂O/10% D₂O and used to assign the spin systems corresponding to the peptide resonances[61–63]. For the 2D-NOESY experiments, mixing times of 300, 150, and 80 ms were acquired to minimize the impact of spin-diffusion in the assignments. Spin-locking fields of 8 kHz and 50 ms mixing time was used for the 2D-TOCSY experiments. All 2D spectral widths were 8000 Hz. The data size was 512 points in F1, indirect dimension, and 2048 points in F2, direct dimension. For each F1 value, 48 transients were accumulated in the NOESY and 32 in the TOCSY experiments respectively. Data were processed with a combination of exponential and shifted sine-bell window functions for each dimension followed by automated baseline- and phase correction using *TopSpin* 3.5 (© Bruker 2020). The 512 × 2k data matrices were zero-filled to 2k × 2k (NOESY and TOCSY). Structure Calculation: To prove that the primary structure (composition and connectivity) corresponds to the theoretical sequence, we have followed the sequence assignment strategy[63]. We identified the characteristic spin system of every residue in the sequence, using the 2D-TOCSY experiment, and every spin system was connected to the following one via NOEs observed from the side chain of a given residue (*i*) to the amide proton of the following residue (*i + 1*) as well as from the amide proton of (*i*) to the amide of (*i + 1*). The full spin analysis as well as the assignment of the NOEs were carried out manually using CARA software[64,65]. Distance restraints derived from the NOESY experiments were used for the NMR-based model building of the peptide in solution using unambiguously assigned peaks exclusively and the program CNS 1.2 (Crystallography and NMR system)[66]. The protocol consisted of an implicit water simulated-annealing of 120 structures using 8,000 cooling steps followed by an explicit water refinement of the calculated structures using all experimental restraints during 1200 steps. The Pd coordination was not explicitly included in the calculation. To display the metal coordination, we manually optimized the His rotamers to facilitate the coordination and the Pd was added to the final model displayed in Fig. 2.

**Computational methods**. Ligand-protein docking exploration with GOLD[30], and GaudiMM[28], was first performed to discern the possible coordination modes of the Pd(II) complex with the **αH-His₂** peptide. Then, the **αH-His₂**[Pd] cores were parametrized with the Amber tool MCPB.py[67], based on the bonded model approach. QM calculations were performed with the DFT formalism as implemented in Gaussian 09, combined with the hybrid functional B3LYP[68]. The basis set employed was 6-31 g(d,p) for C, H and N atoms, while the SDD basis set with *f*-polarization function was used for the metal ion, with a pseudo-potential for the core electrons[69]. All these calculations were combined with the Grimme's D3 correction for dispersion[70]. The complexes were embedded in a solvent-polarizable dielectric continuum model (SMD, water)[71]. Once these QM calculations were obtained, the parameters were created for those atoms coordinating the Pd(II) ion through Seminario's approach[72]. The point charges were computed based on the RESP protocol[73]. Once the parameters were obtained, the systems were set up with AMBER18 tleap to build the topology and the coordinates file. The force field used for the protein atoms was the ff14SB[74], improved force field recommended for the simulation of peptides. For the remaining atoms, the GAFF force field was applied; while the parameters obtained from the MCPB.py calculation were used for the Pd(II) ion. Explicit water TIP3P solvent was also considered in the complexes[75], introducing neutrality by the addition of three and one chloride ions for the metallic and free peptide, respectively. The systems were embedded into a cubic box containing between 2700 and 3000 water molecules, 10 Å from the protein to the edge of the box. The topology and the coordinates files achieved for each complex were then used for a first 10 ns classical Molecular Dynamics (MD) simulation. The coordinates of the last structure were used for the Gaussian accelerated Molecular Dynamics (GaMD) simulations[76], which allows an extensive

exploration of the conformational space. The AMBER ff14SB force field in an NVT ensemble was also used with SHAKE algorithm. A boost on both dihedral a total potential energy was applied, producing GaMDs of at least 2 μs for each system. To prove the correct exploration of the conformational space, PCA analysis were applied to the trajectories. Despite GaMD allows extensive conformational exploration, simulations were started from different putative folded geometries of the peptide to assure unbiased predictions to evaluate the progression of the system from the two extreme situations (data not shown).

**Protein Expression and Purification.** The constructs *pDNA2.0 6H-TEV-KRAS, pDNA2.0 6H-TEV-KRAS G12V* and *pDNA2.0 6H-FLAG-TEV-KRAS G12C* were purchased from *Addgene*. BL21 cells (*Invitrogen*) were transform with the constructs and grown in Luria broth (LB) to OD 600 0.7 and induced with 250 mM isopropyl β-D-1-thiogalactopyranoside (IPTG) for 16 h at 16 °C. Cells were pelleted and resuspended in lysis buffer (20 mM sodium phosphate, pH 8.0, 500 mM NaCl, 10 mM imidazole, 1 mM 2-mercaptoethanol (BME), 5% (vol/vol) glycerol). After centrifugation, the pellet was frozen until use. Then, protease inhibitor (EDTA free) and lysozyme (1 mg/mL) were added, and the pellet was sonicated for 6 min. (35 sec. on /10 sec. off). Protein was purified over an IMAC (immobilized metal affinity chromatography) following *HisPur* Ni-NTA protocols (*Thermo Fisher*). Protein was concentrated in a 10-kDa Amicon Ultra-15 (*Millipore*) aliquoted with 15% glycerol, and then flash-frozen and stored at −20 °C.

**Nucleotide Release assays.** Loading of fluorescent-labeled GDP (mantGTP) to KRAS$^{wt}$ was conducted following previous reports[8,77]. Purified KRAS$^{wt}$ was buffer-exchanged in *Zeba* Spin Desalting Columns (*Thermo Fisher*) in loading buffer (20 mM Tris-HCl [pH 7.5], 50 mM NaCl, 4 mM EDTA and 1 mM DTT). The resulted eluted KRAS$^{wt}$, after measuring their concentration, was incubated with 10-fold molar excess of mantGTP (*Abcam*) for 1.5 h at 20 °C in the dark. Reactions were supplemented with 10 mM MgCl$_2$ and incubated for 1 h on ice. Free nucleotide was removed using a *Zeba* Spin Desalting Columns (*Thermo Fisher*) that had been equilibrated with the reaction buffer (20 mM Tris-HCl, pH 7.5, 50 mM NaCl, 1 mM MgCl$_2$ and 1 mM DTT). The effect of peptides on the intrinsic rate of nucleotide release was monitored using the decrease in fluorescence with time as mantGTP dissociates from KRAS in a 100 μL reaction mixture (96-well plate) of 1 μM KRAS$^{wt}$/mantGPT complex and 10 μM of **αH-His$_2$** or **αH-His$_2$[Pd]**. Fluorescence was excited at 370 nm and emission was monitored at 430 nm during 30 min, using a *Tecan* Infinite M Plex plate reader. The data were processed using the program *GraphPad Prism*.

**Cell internalization and inhibition of ERK1/2.** All steps were performed on a sterile clean bench Telstar AV-100 at rt. Solutions stored in a fridge were warmed beforehand in a water bath (37 °C). Unless otherwise specified, all incubations were performed in DMEM containing 5% of fetal bovine serum (FBS-DMEM). Cell Culture: A549 cell line was cultured in DMEM (Dulbecco´s modified Eagle´s medium), 5 mM glutamine, penicillin (100 units/mL) and streptomycin (100 units/ mL), all from *Invitrogen*. Proliferating cultures were maintained in a 5% CO$_2$ humidified incubator at 37 °C. For all the experiments, cells were seeded in the corresponding well at the indicated concentration 2 days before treatment. A549 cells were seeded on glass-bottom plates 48 h (150.000 cell/ml) before treatment. Culture medium was removed and DMEM containing 5% fetal bovine serum (FBS-DMEM) and **αH-His$_2$** or **αH-His$_2$[Pd]** (10 μM) were added. The metallopeptide was made just before the addition to cells by pre-incubating with metal complexes (1:1) in water for 10 min. After 30 min of incubation with cells, these were washed twice with PBS and replaced with fresh FBS-DMEM to observe under the microscope with adequate filters. Digital pictures of the different samples were taken under identical conditions of gain and exposure. A549 cells were seeded on glass-bottom plates (150.000 cell/ml) 48 h before treatment. Culture medium was removed, the cells was serum-starved and **αH-His$_2$**, **αH-His$_2$[Pd]** or *cis*-PdCl$_2$(en) were added at different concentrations and incubated for 4 h. After two washes with PBS, the cells were lysed in Laemmli buffer containing Tris pH 6,8 1 M, Glycerol, SDS, Bromophenol Blue, and β-mercaptoethanol, and heated at 95 °C for 10 min. The samples were separated by SDS-12.5% polyacrylamide gel and transferred to nitrocellulose membrane, to probed by Western blotting. Levels of total ERK2 and phosphorylated ERK were detected with anti-ERK1/2 and phos-pho-ERK1/2) antibodies, respectively and revealed with Luminata™ Classico Western HRP substrate following manufacter´s protocol. The visualization was performed with ChemiDoc MP imaging system by *Biorad*, and the processing of the images and densitometry quantification was carried out with ImageLab program.

**Fluorescence microscopy.** All images were obtained with an *Andor* Zyla mounted on a *Nikon TiE*. LED excitation wavelength 550 nm. Filter cube TRITC-B-000 (Semrock): BP 543/22 nm, LP 593/40 nm, and DM 562 nm. Images were further processed with *Image J*.

**Reporting summary.** Further information on research design is available in the Nature Research Reporting Summary linked to this article.

## Data availability
The PDB structures of the NMR and Molecular Dynamics studies are available as Supplementary Data 1; the original WB and microscope images are available as Supplementary Data 2. All other datasets generated and analyzed during the current study are available from the corresponding authors on reasonable request.

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

## Acknowledgements

Financial support from the Spanish grants RTI2018-099877-B-I00, CTQ2015-70698-R, Orfeo-cinqa network CTQ2016-81797-REDC, Red the Péptidos en Biomedicina y Nanociencia, RED2018-102417-T; Centro Singular de Investigación de Galicia accreditation 2019-2022, ED431G 2019/03 and the European Union (European Regional Development Fund - ERDF) and AGAUR (SGR-50); the *European Research Council* (Advanced Grant No. 340055), are gratefully acknowledged. S. Learte-Aymamí thanks the MINECO for her Ph.D. fellowship. G. Sciortino thanks the Spanish MICINN Juan de la Cierva program, FJC2019-039135-I. We also acknowledge institutional funding from

MINECO through the Centers of Excellence *Severo Ochoa* Award given to *IRB Barcelona*, as well as from the CERCA Program of the Generalitat de Catalunya. M.J.M. is an ICREA Programme Investigator. Molecular graphics performed with UCSF Chimera, developed by the *Resource for Biocomputing, Visualization, and Informatics* at the UCSF, with support from NIH P41-GM103311[78]. We also want to thank Dr. Huy Hoang and Prof. David P. Fairlie, from The University of Queensland, Brisbane, for sharing the NMR structure of the peptide Ac-[KAAAD]-NH$_2$ used in our preliminary modeling. Finally, we also thank R. Menaya-Vargas for her technical assistance with cell experiments.

## Author contributions

Conceptualization, M.E.V. and J.L.M; methodology, J.R.C., J.-D.M., M.J.M., J.L.M., and M.E.V.; Validation, S.L.A., P.M.M., L.R.M., G.S., J.R.C., J.-D.M., M.J.M., J.L.M., and M.E.V.; formal analysis, S.L.A., P.M.M., L.R.M., G.S., J.R.C., J.-D.M, and M.J.M.; investigation, S.L.A., P.M.M., M.J.M., L.R.M. and G.S.; resources, M.E.V., J.L.M., M.J.M. and J.-D.M.; data curation, S.L.A., P.M.M., L.R.M., G.S. and J.R.C.; writing—original draft, M.E.V., J.L.M., M.J.M. and J.-D.M.; writing—review and editing, S.L.A., P.M.M., L.R.M., G.S., J.R.C., J.-D.M., M.J.M., J.L.M., and M.E.V.; visualization, J.R.C., J.-D.M, M.J.M., J.L.M., and M.E.V.; supervision, J.R.C., J.-D.M., M.J.M., J.L.M., and M.E.V.; project administration, M.E.V.; funding acquisition, M.E.V., J.L.M., M.J.M. and J.-D.M. Author Contributions are detailed as Supplementary Note 1 in the Supplementary Information file.

## Competing interests

The authors declare no competing interests.
