## [Peer Review File · Communications Chemistry]

Reviewers' comments:

Reviewer #1 (Remarks to the Author):

The article emerged for us a unique metallopeptide, which presented a powerful function by detailedly mechanical experiments. The Pd (II) played a vital role for forming stable α H-His2 peptide and affiniting for KRAS. Metallopeptide could also modulate a signaling pathway in living cells. It is potential for developing metalpeptides in drug dicoverly. I consider this to be important work, well suited for publication in communication chemistry. However, the following issue should be considered :

Have authors conducted experiments to screen other transition metals? Perhaps, the nature of metals needs to take into consideration. And owing to the harm of Pd(II) for organism, it is an important issue for employing other metals such as iron which could construct metalpeptides.

Reviewer #2 (Remarks to the Author):

Inhibition of K-Ras-SOS1 interaction is an effective strategy to impair the Ras signaling activity and can provide potential modulators for K-Ras dependent cancer therapy. The authors developed a bis-histidine peptide derived from SOS1, which can form alfa-helix conformation upon coordination with Pd(II). This Pd(II)-binding cyclopeptide (α H-His2[Pd]) displayed a high binding affinity of 345 nM and 294 nM with different K-Ras proteins and inhibited K-Raswt GTP exchange. Besides, α H-His2[Pd] could penetrate into cells and impaired the Ras signaling activity. It is a piece of elegant work and provides a new candidate for Ras inhibitors.

But, I still have some concerns as below.

1) How about the stability of α H-His2[Pd] in the physiological conditions, including the coordination of peptide with metal in the presence of thiol reagent or other complex biomolecules and peptide's degradation curve in serum? It is necessary to test these properties of α H-His2[Pd].

2) The effective concentrations of α H-His2[Pd] for inhibiting Ras signaling activity are tens of micromolar (μ M) in the cell assays. In contrast, its binding affinity with K-Ras obtained from in vitro fluorescence anisotropy assay is around hundreds of nM. The off-target effects should be considered and need to be addressed. I am also concerning about the toxicity of Pd(II) after peptide degradation in cells.

3) I suggest to measure the cell growth inhibition of α H-His2[Pd]. The dependence of cell grow inhibition on the decreasing of Ras signaling activity upon treatment of α H-His2[Pd] should be tested.

4) The cell-penetrating capability of α H-His2[Pd] was characterized by fluorescence microscopy. It is necessary to analyze its specific cell localization. After its binding with K-Ras, was it tethered to membrane as well?

5) In addition, P values should be determined in Figure 4 and Figure 5.

Santiago de Compostela, March 29th, 2022

Dear Reviewers,

We sincerely thank you for your constructive comments and suggestions, which we believe have helped us build a more robust paper. Below you will find a point-by-point answer to your concerns detailing the corrections and additions in the updated version of the manuscript and the Supporting Information.

Responses to the Reviewers:

Reviewer #1 (Remarks to the Author):

The article emerged for us a unique metallopeptide, which presented a powerful function by detailedly mechanical experiments. The Pd (II) played a vital role for forming stable α H-His2 peptide and affiniting for KRAS. Metallopeptide could also modulate a signaling pathway in living cells. It is potential for developing metalpeptides in drug dicoverly. I consider this to be important work, well suited for spublication in communication chemistry. However, the following issue should be considered :

We thank the Reviewer for considering this as “*important work, well suited for publication in communication chemistry*”. We are of course excited about the prospect of having our manuscript published in this journal.

Have authors conducted experiments to screen other transition metals? Perhaps, the nature of metals needs to take into consideration. And owing to the harm of Pd(II) for organism, it is an important issue for employing other metals such as iron which could construct metalpeptides.

The Reviewer’s intuition about the relevance of the metal ion is correct. Earlier experiments in our laboratory described in S. Learte-Aymamí, N. Curado, J. Rodríguez, M. E. Vázquez, J. L. Mascareñas, Metal-Dependent DNA Recognition and Cell Internalization of Designed, Basic Peptides. *J. Am. Chem. Soc.* **139**, 16188–16193 (2017), show that square planar *cis*-Pd(II) complexes present the ideal coordination properties to achieve an effective coordination to the histidines, and facilitate the helical folding. Other metal ions, such as Cu(II), Co(II), Ni(II), Fe(II), did not have this effect. We have included a short comment about the relevance of the nature of the metal ions in the revised manuscript (pages 3-4).

Reviewer #2 (Remarks to the Author):

Inhibition of K-Ras-SOS1 interaction is an effective strategy to impair the Ras signaling activity and can provide potential modulators for K-Ras dependent cancer therapy. The authors developed a bis-histidine peptide derived from SOS1, which can form alpha-helix conformation upon coordination with Pd(II). This Pd(II)-binding cyclopeptide (α H-His₂[Pd]) displayed a high binding affinity of 345 nM and 294 nM with different K-Ras proteins and inhibited K-Raswt GTP exchange. Besides, α H-His₂[Pd] could penetrate into cells and impaired the Ras signaling activity. It is a piece of elegant work and provides a new candidate for Ras inhibitors.

We thank the Reviewer for their comments and for considering this manuscript “*a piece of elegant work and provides a new candidate for Ras inhibitors*”, as well as for the important control experiments suggested in the report and discussed below.

But, I still have some concerns as below.

1) How about the stability of α H-His₂[Pd] in the physiological conditions, including the coordination of peptide with metal in the presence of thiol reagent or other complex biomolecules and peptide's degradation curve in serum? It is necessary to test these properties of α H-His₂[Pd].

We have performed the required experiments, which confirmed the stability of the metallopeptide in PBS, DMEM and Hela cells lysates under similar conditions to those of the cellular assays (4 h at 37 °C). This information has been included in Figure S7, page S16 of the Supporting Information.

2) The effective concentrations of α H-His₂[Pd] for inhibiting Ras signaling activity are tens of micromolar (μ M) in the cell assays. In contrast, its binding affinity with K-Ras obtained from in vitro fluorescence anisotropy assay is around hundreds of nM. The off-target effects should be considered and need to be addressed.

Although the concentration in the incubation medium for the cellular assays is much higher than that required for KRAS binding, the actual intracellular concentration of the metallopeptide α H-His₂[Pd] will be clearly lower. To determine the amount of internalized metallopeptide, we performed ICP-MS analysis of cells incubated in the same conditions used in the ERK inhibition assay. As shown in Figure S8 (page S17 in the updated Supporting Information), only a tiny percentage of the metallopeptide can enter the cells. We measured an uptake of about 0.6% upon incubation with 10 μ M and about 0.4% at higher concentrations (50 μ M); the lower % uptake at higher concentrations reflects the limited uptake capacity of the cells.

I am also concerning about the toxicity of Pd(II) after peptide degradation in cells.

To analyze the toxicity of the Pd(II) complexes we performed MTT assays at long incubation times. The result did not show any significant change in cell viability, which remains high even upon incubation with 100 μ M of the Pd precursor (Figure S9, page S17 in the updated Supporting Information).

3) I suggest to measure the cell growth inhibition of α H-His₂[Pd]. The dependence of cell growth inhibition on the decreasing of Ras signaling activity upon treatment of α H-His₂[Pd] should be tested.

The MTT assays showed a moderate decrease on cell growth at high concentrations of metallopeptide. Although, whether this is due to KRAS inhibition cannot be determined by this assay, this reduction is compatible with our hypothesis and consistent with the results obtained in the rest of experiments performed in cells.

4) The cell-penetrating capability of α H-His₂[Pd] was characterized by fluorescence microscopy. It is necessary to analyze its specific cell localization. After its binding with K-Ras, was it tethered to membrane as well?

Fluorescence microscopy analysis of cells incubated with α H-His₂[Pd] showed mainly endosomal localization. This analysis also showed that the internalization by endocytosis of α H-His₂[Pd] is superior to that of the apo-peptide. It is extremely difficult to know where the peptide is located after binding to KRAS because the concentration of the complex would be very low and also because of the high background emission of the endosomal metallopeptide. Nonetheless, the ERK1/2 inhibition experiments support this hypothesis that part of the metallopeptide is available for binding to RAS.

5) In addition, P values should be determined in Figure 4 and Figure 5.

We have performed statistical analysis in those experiments: Data were subjected to a One-Way ANOVA statistical analysis. Figure 4 and Figure 5 in the manuscript have been updated accordingly.

We thank the Reviewers, and we hope that they find this updated version that incorporates their suggestions appropriate for publication in *Communications Chemistry*.

Sincerely,

A/Prof. M. Eugenio Vázquez

REVIEWERS' COMMENTS:

Reviewer #1 (Remarks to the Author):

The authors have addressed the comments quite well. The reviewer suggests to accept the article.

Reviewer #2 (Remarks to the Author):

The authors provided the results of newly performed experiments and explanations. Most of the concerns have been well addressed. The authors might demonstrate the dependence of cell growth inhibition on the decreasing of Ras signaling activity in their future work.

I agree to accept it as its current state.